# Pre-Calc: Learning to Use the Calculator Improves Numeracy in Language Models

**Vishruth Veerendranath** [1]   **Vishwa Shah** [1]   **Kshitish Ghate** [1]

## Abstract

Quantitative and numerical comprehension in language is an important task in many fields like education and finance, but still remains a challenging task for language models. While tool and calculator usage has shown to be helpful to improve mathematical reasoning in large pretrained decoder-only language models, this remains unexplored for smaller language models with encoders. In this paper, we propose **Pre-Calc**, a simple pre-finetuning objective of learning to use the calculator for both encoder-only and encoder-decoder architectures, formulated as a discriminative and generative task respectively. We pretrain BERT and RoBERTa for discriminative calculator use and Flan-T5 for generative calculator use on the MAWPS, SVAMP, and AsDiv-A datasets, which improves performance on downstream tasks that require numerical understanding. Our code and data are available at https://github.com/calc-cmu/pre-calc.

## 1. Introduction

The advancement of language modeling in natural language processing has significantly impacted various computational tasks. However, the intricacy of numerical and quantitative comprehension in text remains a challenging frontier. Numerals, unlike words, possess unique characteristics that necessitate specialized handling by language models either in tokenization or processing. This necessity becomes particularly evident in tasks involving quantitative reasoning, where the ability to interpret and manipulate numerical information is crucial.

Numeracy involves majorly 2 properties. The first is *semantic reasoning* which focuses more on the understanding of relations in text and the second is *computational ability*

[1]Carnegie Mellon University. Correspondence to: Vishruth Veerendranath <vveerend@andrew.cmu.edu>.

*The First AI for MATH Workshop at the 41st International Conference on Machine Learning*, Vienna, Austria. Copyright 2024 by the author(s).

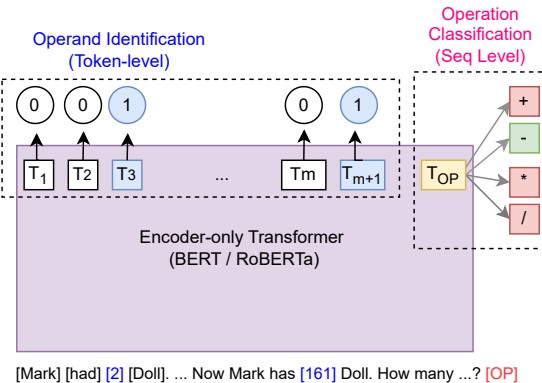

*Figure 1.* Pre-Calc for Encoder-Only models

which focuses on performing explicit mathematical operations. Hence, the aim is to develop systems that can perform explicit mathematical operations while retaining or improving its quantitative reasoning.

We present **Pre-Calc**, a pre-finetuning objective of learning to use the calculator, to improve numerical abilities in language models. We propose Pre-Calc objectives for both the encoder-only and encoder-decoder classes of language models, and use a combination of the MAWPS (Koncel-Kedziorski et al., 2016), SVAMP (Patel et al., 2021), AsDiv-A (Miao et al., 2020) datasets to pre-finetune the models.

Our encoder-only objective, used to pre-finetune BERT and RoBERTa models, offers quick and efficient processing suitable for tasks where speed is paramount. The Pre-Calc versions of the models show competent performance on 6 quantitative downstream tasks from Chen et al. (2023) and substantial improvements on 4 sub-tasks with an improvement greater than 10 points for RedditNLI and AWPNLI specifically.

Similarly, our encoder-decoder approach, used to pre-finetune Flan-T5, demonstrates an improved ability to perform explicit computations in computation-intensive tasks like AWPNLI. Although there is a noted trade-off, with a slight decrease in performance on text-focused and semantic tasks, the objective showcases strengths in processing mathematically intensive language.

Our study underscores the potential of tailored language models to significantly enhance numeracy in NLP, providing an avenue for more efficient and effective processing of numerical data in language.

## 2. Task and Data

### 2.1. Downstream Tasks

We focus on downstream quantitative reasoning tasks, specifically QNLI and QQA (Chen et al., 2023).

**QNLI** is the task of making *natural language inferences* based on quantitative clues. This dataset is adopted from the EQUATE(Ravichander et al., 2019) and is composed of NewsNLI, RedditNLI, AWPNLI and RTE-Quant. StressTest involves numerical reasoning instances from Naik et al. (2018), used as a synthetic sanity check.

**QQA** involves the task of *multiple-choice question answering* involving commonsense as well as quantitative comparisons. The dataset for this is adopted from Task 3 of NumGLUE (Mishra et al., 2022) and the Quarel dataset (Tafjord et al., 2019), which includes questions from quantitative domains such as physics and economics.

### 2.2. Pre-Finetuning Data

We use the MAWPS dataset (Koncel-Kedziorski et al., 2016), SVAMP (Patel et al., 2021) and AsDiv-A (Miao et al., 2020) as the numerical domain datasets for pre-finetuning. These consist of simple arithmetic word problems, along with their numerical solutions. The three datasets create a dataset with **4,225** total examples which are challenging and require understanding the context of numbers, represented either as digits or in words.

We construct this dataset from the Calc-X collection (Kadlčík et al., 2023) that has been annotated with equations for each problem, as well as `<gadget>` annotations in the answer to train a model to use a calculator when the `<gadget>` token is produced. We use the annotations of the equations particularly in our methodology.

## 3. Pre-Calc Methodology

We posit that learning to use a calculator requires understanding of numbers and ways in which numbers can be combined. This is used to formulate the **Pre-Calc** objectives described below.

### 3.1. Encoder-Only

#### 3.1.1. DATA PREPROCESSING

We preprocess Calc-MAWPS, Calc-SVAMP and Calc-AsDiv-A (from the Calc-X collection) (Kadlčík et al., 2023)

and add 2 new features required for Pre-Calc. First is the *operand tag sequence*, which is a sequence of binary tags that is 1 if the original token it corresponds to is an *operand* and 0 if it isn't. Secondly we extract the *Operation*, which is the operation among {+ (add), - (subtract), * (multiply), / (divide)} that is required for the question. We extract the operation either directly from the equation or the reasoning chain in Calc-X and generate the operand tag sequence, by first extracting the operands and then tagging the occurances of the operands in the binary sequence with a 1. As part of this process we also filter out instances where there are more than one distinct operations as part of the equation.

#### 3.1.2. PRE-CALC METHOD

An illustration of the Pre-Calc method for Encoder-only model can be seen in Fig 1. This is decomposed into two tasks as a dual-objective.

Firstly, we use the pretrained Encoder-only language model for the task of *Operand Identification*, which is a token-level classification task. The tags possible for each token are 1 and 0.

Secondly, we perform the task of *Operation Classficiation* by adding a special `[OP]` token at the end of each sequence and using this `[OP]` token's final layer representation to classify the operation required in this sequence (+, -, *, /). Hence, this is essentially a sequence-level classification task similar to classifying from the representation of a `[CLS]` token. However, we do not use the `[CLS]` token at the start of the sequence, to enable this objective even in non-bidirectional models with an autoregressive attention mask (like decoder-only models).

In essence, we use two heads — one token classification head for Operand Identification, and one sequence classification head for Operation Classification — to train it with the dual objective as per Equation 1

$$\mathcal{L} = \mathcal{L}_{operation} + \lambda \mathcal{L}_{operand} \qquad (1)$$

where $\mathcal{L}_{operation}$ is the cross-entropy loss for the sequence classification (`[OP]`) head and $\mathcal{L}_{operand}$ is the binary cross entropy (BCE) loss for the token classification head. Here we empirically set $\lambda = 1$.

#### 3.1.3. DOWNSTREAM TASK INFERENCE

For most downstream tasks, we do not explicitly perform calculator computations using operands and operations predicted by the model and instead use Pre-Calc only as a learning objective before finetuning it for specific downstream tasks. However, as AWPNLI task requires the model to be able to perform calculations explicitly, we utilize an alternative strategy for its inference adapted from our pre-finetuning strategy shown in Fig 1. We first extract the

operand labels for each token from the premise $T_i$ and operation using the $T_{op}$ token. This gives us the operands and operation, after which we automate the calculation of the final answer comparing it with the hypothesis. This helps the model focus on the semantic extraction of operation and offload explicit computation to the calculator.

## 3.2. Encoder-Decoder

Encoder-Decoder or Decoder based models provide the abilities of long-form unbounded generations. This is advantageous for numerical problems, where multiple intermediate operations might be required for computation or reasoning(Wei et al., 2022). By reframing our task to output expressions, we distil the task to output the set of operations, leaving the computation part to the tool.

### 3.2.1. DATA PREPROCESSING

As mentioned earlier, we use the MAWPS dataset for training our model to output expressions. As each instance in MAWPS consists of a question and a single numerical answer, to obtain closer resemblance to NLI format tasks, we reframe the question-numerical answer instance to a pair of complete sentences using prompting with LLaMa-7B (Touvron et al., 2023) as this is a simple text generation task. To obtain contradiction pairs, we perturb the true numerical answer by a small value (ranging from -5 to +5) before passing to the LLaMa-7B model as these will create harder instances for the model to learn from. We additionally also use the Multi-NLI(Williams et al., 2018) dataset to retain and improve textual inferential abilities of the model. We train on this combined data as Seq2Seq generation task. Combining these tasks should allow the model to infer both semantic and computational capabilities.

### 3.2.2. PRE-CALC METHOD

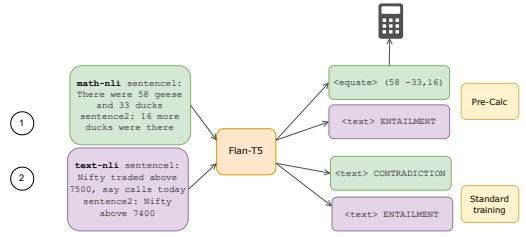

*Figure 2.* Our Encoder-Decoder based approach

We utilize this ability of Seq2Seq modeling by fine-tuning Flan-T5 on our NLI-based tasks for pre-finetuning mentioned in 2.2. As shown in Figure 2, we use a **math-nli** prefix tag for tasks that require mathematical computation eg: MAWPS reformatted as above and use a **text-nli** tag for text-based tasks eg: Multi-NLI. This lets the model decipher whether the task requires explicit calculation - in

which case it should output an expression for tool use, or use its inherent text-capabilities to reason over text numeracy.

Similar to Kadlčík et al. (2023), we make the model output token **<equate>** with the corresponding expression for computational tasks as essentially the task involves equating expressions in sentence 1 and sentence 2 and **<text>** for more textual-numeracy tasks with the final answer. This helps at inference to verify if the final computation requires to go through the calculator or not. We also hope to expand to **<compare>,<compute>** tokens as we extend this method in future to more down-stream tasks. We denote this as our Pre-Calc method for Encoder-Decoder models which performs tool-based pre-finetuning. As our baseline we also evaluate the performance of doing only text-based fine-tuning which we call our Standard Training approach.

## 4. Experiments

### 4.1. Baselines

We compare the performance of our method against several baseline models on tasks that require numeracy. Following Chen et al. (2023), the baseline methods involve reframing techniques, namely *Original*, *Digit-based*, and *Scientific* Notation methods, and are pre-finetuned on the *Comparing Numbers Dataset (CND)*. Each of these methods are applied to both BERT (Devlin et al., 2019) and RoBERTa (Liu et al., 2019) to create two versions of each baseline method.

### 4.2. Encoder-Only

We use the pretrained BERT and RoBERTa base models and pre-finetune as per Section 3.1.2. We use the 4-class cross-entropy loss for training the *Operation Classification* head, and a 2-class cross-entropy (equivalent to binary cross entropy) loss for the *Operand Identification* head. The models are trained with the Adam optimizer for 20 epochs, a batch size of 8, and a learning rate of 5e-4. The checkpoint after this pre-finetuning is named Pre-Calc-BERT or Pre-Calc-RoBERTa[1].

We then finetune Pre-Calc-BERT and Pre-Calc-RoBERTa on the downstream tasks of QNLI and QQA using the same hyperparameters used by the CN-BERT baselines (Chen et al., 2023) — AdamW optimizer with a learning rate of 5e-5, batch size of 8 for 5 epochs. We use 10-fold cross validation to report our results for the tasks where an explicit test split is not available.

### 4.3. Encoder-Decoder

We use Flan-T5 as our base model, For pre-finetuning, we collect a balanced sample consisting about 4200 instances created from MAWPS for math-nli task and 3900 instances extracted from Multi-NLI for text-nli task[1].

| Model | Notation | QNLI | | | | | QQA |
| | | RTE-QUANT | News | Reddit | AWPNLI | Stress Test | |
|---|---|---|---|---|---|---|---|
| BERT | Org | 66.73 | 74.22 | 62.40 | 59.20 | 99.46 | **56.79** |
| | Digit | 60.22 | 75.94 | 62.86 | 53.20 | **99.70** | 52.63 |
| | Sci | 66.80 | 75.60 | 65.14 | 60.73 | 99.46 | 53.33 |
| | CN-Digit | 62.88 | 76.97 | 68.57 | 60.27 | 99.58 | 53.60 |
| | CN-Sci | 66.87 | **77.98** | 65.64 | 54.70 | 99.58 | 52.38 |
| | **Pre-Calc (ours)** | **67.00** | 76.54 | **76.00** | **68.97** | 99.47 | 53.93 |
| RoBERTa | Org | 62.79 | 78.35 | 59.33 | 57.64 | **100.00** | 52.27 |
| | Digit | 62.67 | 79.38 | 63.71 | 56.69 | 99.94 | 58.94 |
| | Sci | 62.93 | 79.37 | 62.88 | 57.41 | **100.00** | 56.47 |
| | CN-Digit | 68.13 | 77.66 | 62.99 | **58.80** | **100.00** | 51.21 |
| | CN-Sci | 63.97 | 74.57 | 63.80 | 58.74 | 99.98 | 53.6 |
| | **Pre-Calc (ours)** | **73.90** | **82.21** | **78.00** | 58.17 | **100.00** | **61.05** |

*Table 1.* Micro-F1-Scores (in %) of Pre-Calc trained models as compared to CN (Comparing Numbers) trained and reframing (Digit, Sci) baselines

As this is a sequence generation task, the objective is same as that in CausalLM modeling for next-word prediction. We use AdamW optimizer with a learning rate of 5e-5, batch size of 8 for 5 epochs. We do not perform any fine-tuning on our downstream tasks, and show results for prompt based few-shot evaluations on each task. We call our model FlanT5-Pre-Calc [1] and use 2 baselines, Flan-T5 few-shot and Flan-T5-ST only with standard text training[1].

## 5. Results and Discussion

### 5.1. Encoder-Only

Our evaluation on the QNLI and QQA tasks, as outlined in Table 1, demonstrates the efficacy of our Pre-Calc approach. For BERT, our Pre-Calc method significantly outperforms all other reframing techniques for RedditNLI, AWPNLI and RTE-Quant. These results highlight the effectiveness of our method in dealing with diverse numerical information in natural language. In the case of RoBERTa, the Pre-Calc approach consistently outperformed other methods across all three tasks - RTE-Quant, NewsNLI and RedditNLI. This performance is markedly superior compared to the original RoBERTa and other variants that use the reframing techniques, with lower scores in all categories.

For AWPNLI we report results for baselines from Chen et al. (2023), and for our results we compute F1-score on the complete dataset using our methodology described in section 4.2. We see a substantial improvement in Pre-Calc compared to the earlier baselines which can be attributed to our training and inference strategy which can precisely attend and compute an expression which is essential for the AWPNLI task.

---

[1] https://huggingface.co/collections/Calc-CMU/pre-calc-657a5ad5f1ae42fb12364563

In QQA as well, Pre-Calc-RoBERTa improves performance over its counterpart. This indicates that Pre-Calc improves commonsense reasoning abilities and this effect is more pronounced in RoBERTa which is a stronger base model.

Overall, the results validate our hypothesis that the Pre-Calc approach, which integrates calculator-like capabilities into the model, significantly enhances performance in tasks requiring numeral-aware semantic and computational capabilities .

### 5.2. Encoder-Decoder

We present the results for our Encoder-Decoder based approach in Table 2. We see that for AWPNLI which requires explicit computation, FlanT5(Pre-Calc) achives almost double performance compared to FlanT5-few shot and FlanT5-ST, showing that Flan-T5 originally did not have this capability to evaluate expressions and compare values and this property cannot be instilled only via text finetuning as can be seen from the performance of FlanT5-ST. Further compared to prior works (Chen et al., 2023), this achieves state-of-the art results on AWPNLI.

However, we see that the performance slightly decreases on NewsNLI and RTEQuant which are more text-focused tasks. We see that original pretrained FlanT5 does better at this as it already has inherent properties to handle semantic numeracy. This is likely because training with specific tasks discussed above causes forgetting/over-fitting in the model. This can also be attributed to the language-modeling MLE loss which focuses more on generating outputs specific to the format discussed in Fig 2 rather than its original properties of in-context learning and reasoning. To combat this, in the future we hope to regularize learning better so that a diversity of tasks can be included avoiding overfitting in the model.

| Model \Task | AWP NLI | News NLI | RTE Quant |
|---|---|---|---|
| Few-shot | 41.56 | 77.47 | **85.74** |
| ST (ours) | 37.55 | **76.75** | 73.43 |
| Pre-Calc (ours) | **80.29** | 75.20 | 71.26 |

*Table 2.* Micro F1-score of Flan-T5-large when using our Encoder-Decoder based approach

## 6. Analysis

### 6.1. Dual-Objective in Encoder-Only Pre-Calc

We inspect the characteristics of the two objectives during pre-finetuning. Fig. 4 shows the F1-score across epochs for the operand identification objective on the validation data. While this seems to fluctuate, it consistently stays above 90% (the accuracy for this task also consistently remains at about 99%), indicating that the operand identification task is not very challenging and that there is very little loss signal from this task beyond the first few epochs. Regardless of having the second objective, the F-1 for this task is still maintained at a high number.

In Fig. 5 we see the accuracy plot on validation data for the operation classification objective across epochs. Here we see that the accuracy consistently increases but still remains under 75% which tells us that this objective is a lot more challenging, which is also explained by the fact that it has to be inferred from text. Together, the two aid different abilities — picking numbers out with operand identification and combining numbers with operation classification — which are both important for any downstream quantitative task

### 6.2. Operation wise difficulty for FlanT5

We sample 500 instances from the MAWPS reframed dataset, to observe operation-wise accuracy for the model. We observe in Figure 3 that about 60% errors are for instances that entail a divide operation. This could be because understanding division requires the model to develop an understanding of what operand should be the numerator and which should be the denominator. There are also rare

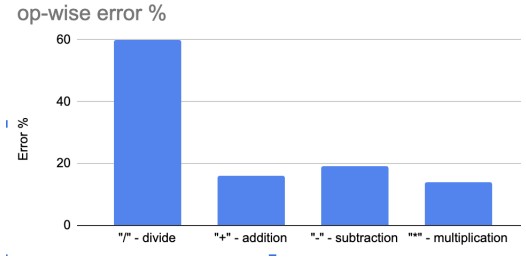

*Figure 3.* Operation wise error for FlanT5-Pre-Calc

instances where the model is required to understand the idea of ratio-proportion which requires more complex understanding compared to other operations.

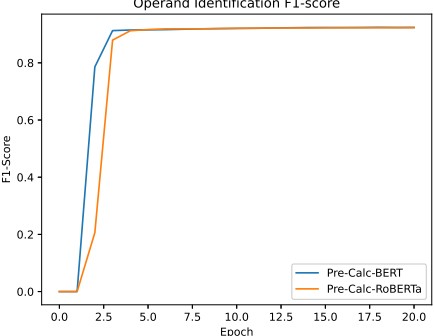

*Figure 4.* Operand Identification Loss Plot

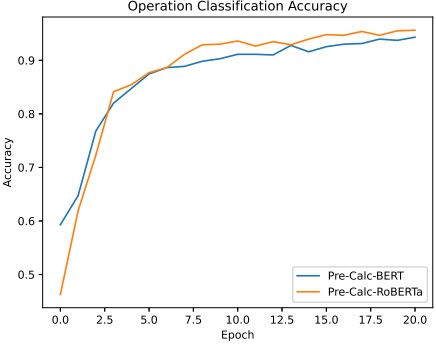

*Figure 5.* Operation Classification Loss Plot

## 7. Related Work

**Numeracy in LMs** Numeracy, or the ability to understand and work with numbers, is a critical aspect that has been relatively underexplored compared to other linguistic competencies in NLP models. Spithourakis & Riedel (2018) emphasized the need for LMs to better understand numbers, setting a precedent for subsequent research.

Chen et al. (2019) introduced Numeracy-600K, a large-scale dataset designed to improve the ability of models to detect exaggerated information in financial texts. Concurrently, Wallace et al. (2019) explored the embedding properties of numbers, shedding light on how numeracy can be integrated into LMs. Zhang et al. (2020) analyzed the representation of numerals in scientific notation, addressing the challenge of scale understanding in LMs. Chen et al. (2021) furthered this exploration by suggesting a digit-based encoder for numeral encoding, providing a novel perspective on numeral representation.

**Pre-Finetuning** In addition to these studies focused on numeral representation, other researchers have investigated the potential of pre-finetuning tasks to enhance LM capabilities. Aghajanyan et al. (2021) introduced a massive multi-task representation with pre-finetuning, demonstrating the efficacy of pre-finetuning in improving model performance across a range of tasks. Geva et al. (2020) proposed GENBERT, which is trained on automatically-generated synthetic data in a multi-task setup. This training significantly improves performance on numerical reasoning tasks such as DROP and math word problems, while maintaining high performance on standard reading comprehension tasks. Wang et al. (2017) presented a deep neural solver, a hybrid model combining the RNN with a similarity-based retrieval to translate math word problems into equation templates.

**Tool-Use** Gou et al. (2023) presented a series of Tool-integrated Reasoning Agents (ToRA) designed to solve complex mathematical problems by augmenting the model with external computational tools. The training process involves collecting interactive tool-use trajectories and applying imitation learning and output space shaping, showcasing the efficacy of combining natural language reasoning with program-based tool use. Kadlčík et al. (2023) introduced Calc-X, a collection of datasets designed to integrate calculator usage into language model reasoning chains. Calc-X consolidates 300,000 samples from several chain-of-thought tasks requiring arithmetic reasoning. The study demonstrates how Calcformers, models trained on Calc-X, significantly enhance the accuracy of generating correct results by offloading computations to symbolic systems.

## 8. Conclusion and Future Work

In this work, we improve the numeracy in language models on the QNLI and QQA tasks which involve textual and computational quantitative reasoning. We do so by proposing calculator usage as a pre-finetuning task in a discriminative and generative fashion for encoder-only and encoder-decoder models respectively. This improves encoder-only models across various downstream tasks and improves encoder-decoder models on tasks that require explicit computation.

Future work can address the balance between textual understanding and numerical reasoning, by refining regularization strategies to maintain the language model's core strengths while enhancing its computational abilities. Tool-use in encoder-only models could also be extended to more complex tools similar to decoder-only models.

## Acknowledgments

We thank Robert Lo for the helpful discussions.

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
