# OpenReview forum: "Pre-Calc: Learning to Use the Calculator Improves Numeracy in Language Models"
_ICML.cc/2024/Workshop/AI4MATH — ICML 2024 Workshop AI4MATH Poster_

### Official Review · Reviewer_B9Lz · 2024-06-10

**Rating:** 3
**Confidence:** 4

**Summary:**

This paper looks at improving how language models handle numbers, which is a challenge in natural language processing. The authors introduce Pre-Calc, a method to train models to better understand and use numbers. They tested this on both encoder-only models (like BERT and RoBERTa) and encoder-decoder models (like Flan-T5) using datasets like MAWPS, SVAMP, and AsDiv-A. Pre-Calc improved the models' ability to handle numerical tasks, with significant gains in some areas, especially in tasks needing strong math skills. However, there was a slight drop in performance on tasks focused on regular text.

**Questions:**

I have no specific questions.

**Reasons To Accept:**

I do not recommend for acceptance.

**Reasons To Reject:**

The paper should be rejected primarily due to its weak relevance to the topic of the workshop, which focuses on AI for mathematics. The proposed task of improving language models (LLMs) to better handle calculators does not align with the central objectives of AI in mathematics. Furthermore, enhancing existing LLMs like BERT and T5 for calculator usage does not push the boundaries of machine learning or natural language processing. The study does not present much insights that could advance frontier research in these areas.

---

### Official Review · Reviewer_JTfy · 2024-06-11

**Rating:** 7
**Confidence:** 3

**Summary:**

The paper introduces Pre-Calc, a pre-finetuning objective designed to enhance the numerical comprehension of language models by teaching them to use calculators. The approach is applied to both encoder-only and encoder-decoder architectures. The authors pre-train BERT and RoBERTa for discriminative calculator use and Flan-T5 for generative calculator use. The models are evaluated on various datasets, showing improved performance on tasks requiring numerical understanding.

**Questions:**

1. What's the advantage of teaching encoder/encoder-decoder models to use calculator over decoder-only language models?

**Reasons To Accept:**

1. The paper outlines a clear and detailed methodology for both encoder-only and encoder-decoder models, providing a dual-objective approach for the former and a sequence-to-sequence generation task for the latter.
2. The authors provide empirical evidence showing significant improvements in performance on numerical tasks, particularly with the encoder-only models on RedditNLI and AWPNLI sub-tasks.

**Reasons To Reject:**

No clear limitations.

---

### Official Review · Reviewer_jwRy · 2024-06-13

**Rating:** 5
**Confidence:** 4

**Summary:**

1. Constructed a pipeline based on small models to complete mathematical calculation task
2. Proposing a decoupling method to decompose mathematical calculations into two stages: operator identification and result generation.

**Questions:**

1. please show the method how to handle multi-step calculate if one sentence contains compound operations

**Reasons To Accept:**

1. The method is friendly to industrial implementation
2. The entire method provides an innovative idea for complex reasoning based on language models.

**Reasons To Reject:**

1. Perhaps as the capabilities of large language models increase, this method will not be effective for a long time.
2. For very complex multi-step reasoning problems, Encoder-only model handle it with the idea of ​​sequence classification may not a robust approach

---

### Meta-Review · Area_Chair_jLM4 · 2024-06-13

**Recommendation:** Accept (Poster)
**Confidence:** 3

**Metareview:**

The paper proposes to improve language models' math ability by teaching them to use calculators. The proposed method can applied to different models and achieves significant improvements on smaller LMs. Although the paper does not experiment with recent LLMs, and current results do not push the boundaries in this field, it provides an innovative idea for complex reasoning based on language models.

---

### Decision · Program_Chairs · 2024-06-13

Accept (Poster)